# How chronic conditions are understood, experienced and managed within African communities in Europe, North America and Australia: A synthesis of qualitative studies

Ama de-Graft Aikins[1]*, Olutobi Sanuade[2], Leonard Baatiema[3], Kafui Adjaye-Gbewonyo[4], Juliet Addo[5], Charles Agyemang[6]

1 Institute of Advanced Studies, University College London, London, United Kingdom, 2 Department of Population Health Sciences, Spencer Fox Eccles School of Medicine, University of Utah, Salt Lake City, Utah, United States of America, 3 Department of Health Policy, Planning and Management, School of Public Health, University of Ghana, Legon, Accra, Ghana, 4 Faculty of Education, Health and Human Sciences, University of Greenwich, London, United Kingdom, 5 Department of Non-communicable Disease Epidemiology, London School of Hygiene and Tropical Medicine, London, United Kingdom, 6 Department of Public Health, Amsterdam Public Health Research Institute, Amsterdam UMC, University of Amsterdam, Amsterdam, The Netherlands

* a.de-graft-aikins@ucl.ac.uk

**Data Availability Statement:** All relevant data are within the paper.

## Abstract

This review focuses on the lived experiences of chronic conditions among African communities in the Global North, focusing on established immigrant communities as well as recent immigrant, refugee, and asylum-seeking communities. We conducted a systematic and narrative synthesis of qualitative studies published from inception to 2022, following a search from nine databases—MEDLINE, EMBASE, PsycINFO, Web of Science, Social Science Citation Index, Academic Search Complete, CINAHL, SCOPUS and AMED. 39 articles reporting 32 qualitative studies were included in the synthesis. The studies were conducted in 10 countries (Australia, Canada, Denmark, France, Netherlands, Norway, Sweden, Switzerland, United Kingdom, and the United States) and focused on 748 participants from 27 African countries living with eight conditions: type 2 diabetes, hypertension, prostate cancer, sickle cell disease, chronic hepatitis, chronic pain, musculoskeletal orders and mental health conditions. The majority of participants believed chronic conditions to be lifelong, requiring complex interventions. Chronic illness impacted several domains of everyday life—physical, sexual, psycho-emotional, social, and economic. Participants managed their illness using biomedical management, traditional medical treatment and faith-based coping, in isolation or combination. In a number of studies, participants took 'therapeutic journeys'—which involved navigating illness action at home and abroad, with the support of transnational therapy networks. Multi-level barriers to healthcare were reported across the majority of studies: these included individual (changing food habits), social (stigma) and structural (healthcare disparities). We outline methodological and interpretive limitations, such as limited engagement with multi-ethnic and intergenerational differences. However, the studies provide an important insights on a much-ignored area that intersects healthcare for African communities in the Global North and medical pluralism on the continent; they also raise important

**Funding:** This work was funded by a British Academy grant - Grant No: GP1\100143 - awarded to the first and corresponding author.

**Competing interests:** The authors have declared that no competing interests exist.

conceptual, methodological and policy challenges for national health programmes on healthcare disparities.

# 1. Background

The global burden of NCDs, such as diabetes, hypertension and cancers, has a disproportionate impact on African populations on the continent and in the African diaspora [1, 2]. Research on chronic illness experiences among Africans has focused predominantly on continental experiences and health research on minoritized ethnic communities in the Global North has neglected African communities [2–5]. Adapting Malfatti (2020) [4] and Braff and Nelson (2022) [5], we define the Global North as 'countries, located primarily, but not exclusively, in the northern hemisphere that have historically been identified as "the West" or "first world" "developed" or "core" due to perceptions of their relative wealth, higher quality of life, technology, and global dominance"; and the associations between their global dominance and exploits of colonialism.' The countries include the United States, Canada, England, nations of the European Union, and countries in the southern hemisphere including Australia, and New Zealand.

A limited set of studies and reviews report multiple barriers to healthcare for African communities in Europe, North America and other countries in the Global North, including inequitable access to healthcare services, language barriers in service delivery, and (conscious and unconscious) biases in diagnostic and treatment practices [3, 6, 7].

There is yet to be an attempt to comprehensively identify, synthesize and theorize evidence of these lived experiences; of how communities living with chronic conditions make sense of their conditions, experience their conditions and manage treatment and care. Synthesizing the available evidence on chronic illness experiences and care will aid a more nuanced understanding of shared and divergent patterns of lived experiences of chronic conditions within African diaspora communities, as well as responses–such as spiritual, psychological, psychosocial, and political—to living with chronic conditions that provide insights for developing context-specific, responsive and equitable chronic care models.

## 1.1. Aim

The aim of this review was to examine the lived experiences of chronic conditions among African communities in Europe and North America, and other countries in the Global North, focusing on established immigrant communities as well as recent immigrant, refugee and asylum seeking communities. We were interested in meanings, experiences and illness action from the perspective of individuals living with chronic conditions.

Research on the continent and diaspora suggest African communities hold complex chronic illness beliefs that incorporate medical, cultural and common sense ideas, a social psychological phenomenon called 'cognitive polyphasic'—and defined as a process whereby individuals and communities draw on diverse, and even conflicting, models of knowledge to make sense of new phenomena or re-examine old phenomena with or without psychological tension [8]. These cognitive polyphasic processes inform eclectic healthcare practices that blend biomedical, ethnomedical and faith-based interventions [3, 8]. For communities in the Global North beliefs and practices are further mediated by enculturation—"the process of selectively acquiring or retaining elements of one's heritage culture while also selectively acquiring some elements from the receiving cultural context" [9]—and healthcare solutions are sought through transnational therapy networks [10, 11].

We had three questions: how did individuals living with chronic conditions understand their condition(s) (eg. lay theories about causes and nature of conditions)?; how did individuals experience their condition(s) (e.g short-term and long term impact of living with chronic conditions)?; what did individuals do about their condition(s) (how did individuals access and utilize treatment services, social support and care, and what were the pathways to these systems of care)?

## 2. Method

### 2.1. Protocol and registration

This review was conducted and reported in accordance with PRISMA guidelines (S1 Table). The protocol was registered on PROSPERO (CRD42019162565). Since this was a systematic review, ethical approval was not sought.

We used the Preferred Reporting Item for Systematic Reviews and Meta-Analyses (PRISMA) extension for screening and reporting (Fig 1) [12]. We searched 9 electronic databases (MEDLINE, EMBASE, PsycINFO, Web of Science, Social Science Citation Index, Academic Search Complete, CINAHL, SCOPUS and AMED) from inception until June 2022 and hand searched reference lists of relevant papers. The search strategy was developed in consultation with an experienced Librarian. The search strategy is available in S2 Table. Qualitative research articles from the social science and medical grey literature on Google Scholar were also conducted for additional eligible articles. Only studies published in the English language were retrieved.

### 2.2. Selection of studies

Peer-reviewed qualitative studies that explored and reported on the lived experiences and systems of care for chronic conditions for Africa communities in the Global North were included. Studies that used qualitative methods such as interviews, focus group discussions, and observations were included. Mixed methods studies were also included; however, only the qualitative aspects of the study were extracted for analysis. Studies that were quantitative by design or did not report any information relating to lived experiences of chronic conditions or systems of care were excluded. Studies that reported on the lived experiences of patients based on the views of care-givers or health care professionals were also excluded in this review.

The search was carried out by one reviewer. All studies from the databases search process were exported into EndNote for screening. The screening process was carried out in four stages. The first involved the removal of duplicates aided by EndNote. Applying the eligibility criteria, the second stage was based on title relevance. Thirdly, all articles retained after title screening were retained for further screening of abstracts for potential relevance and eligibility —this was conducted by two reviewers. Finally, full articles were retrieved and screened for eligibility. Where there were disagreements in reviewers' decisions to include or exclude a potential study, a third reviewer was consulted.

### 2.3. Data extraction and quality assessment

Data from eligible studies were extracted using a standardized extraction form, guided by the CONSORT checklist [13]. Characteristics of the included studies, including authors, year of publication, study country, study aim, study design, study population types (sex, race, ethnicity, people with chronic condition, caregivers, health care providers or multiple population), methods, type of chronic condition and key findings, were extracted. In addition, 1–2 page summaries were developed for each article focusing on the aforementioned characteristics, as

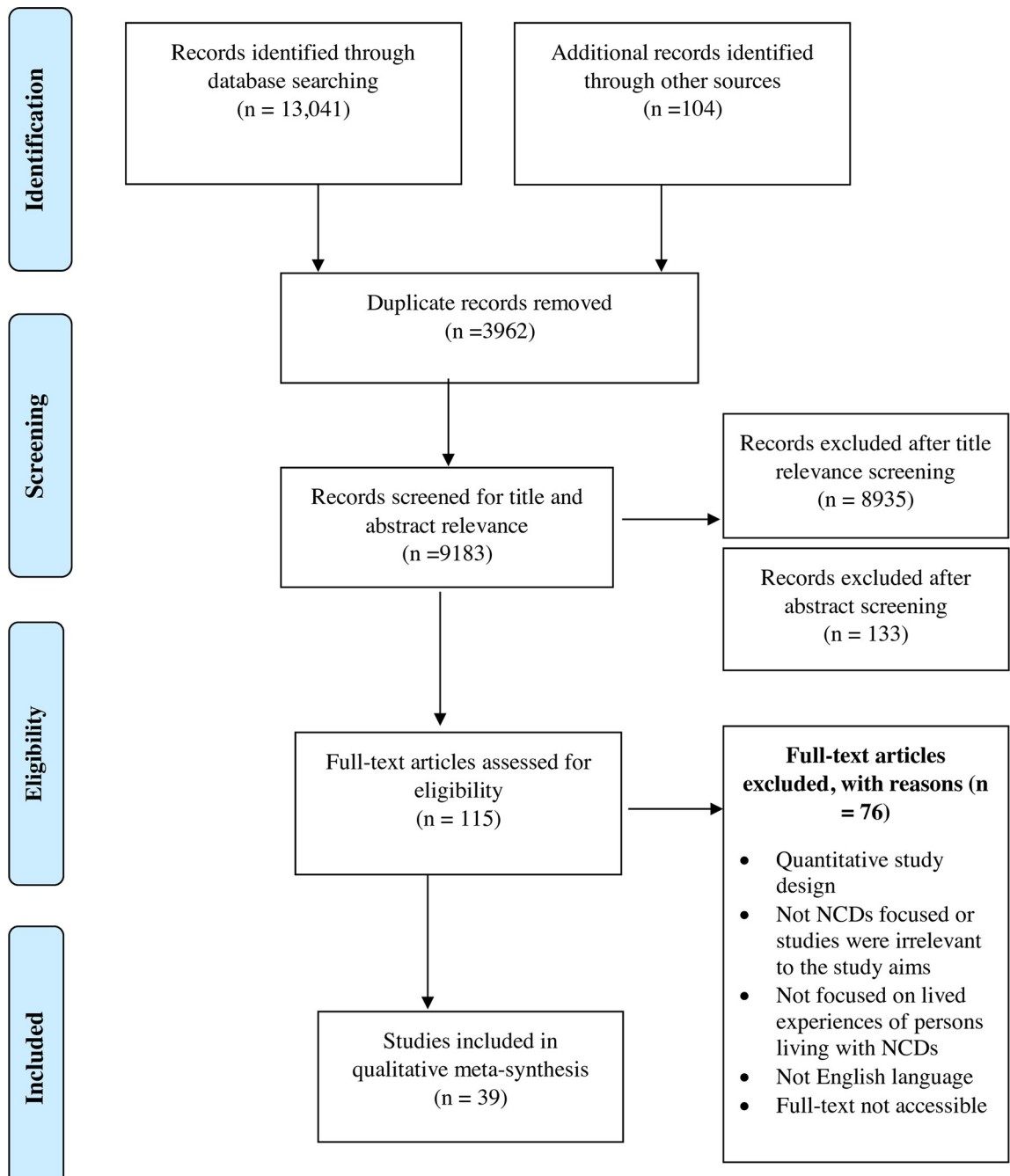

**Fig 1. Flow chart of literature search and inclusion.**

well as institutional affiliation of authors, their disciplines, research questions, conceptual framework, methods and key findings. The Consolidated Criteria for Reporting Qualitative Research (COREQ) was used to assess the quality of reported in all included studies (**S3 Table**) [14]. The 32-item assessment tool was based on 32 questions that examine item appropriateness, validity, and relevance of the studies based on three domains: research team and reflexivity, study design and analysis, and findings. Two reviewers assessed the application of the COREQ 32 items and major discrepancies exist were resolved by a third reviewer.

## 2.4. Data analysis

A three-stage thematic analysis strategy was applied to the summary table and paper summaries. The first stage involved mapping patterns across the thematic areas (e.g study locations, focal communities, chronic conditions) (see Table 1). The second stage analysed participant narratives and author interpretations on the three thematic areas–meanings, experiences and illness action. The third stage focused on conceptual intersections across the studies. Following the meta-ethnography tradition (e,g Campbell et al, 2003 [15])–where the goal is a meta-theory that transcends individual theories used by papers–we expected new conceptual insights to emerge from the selected studies. We paid systematic attention to the concepts used in the studies and how these concepts were operationalised and how they informed interpretation of data. Some studies used concepts that captured shared findings succinctly. Where appropriate we use concepts developed and/or used in the reviewed studies as interpretive guides.

## 3. Results

39 papers reporting 32 qualitative studies met the inclusion criteria [16–54]. 14 papers reported different aspects of 7 studies [17, 18, 23, 24, 26, 27, 30, 31, 38, 39, 47, 48, 52, 53]. Alloh, 2019, 2021; [17, 18] Beune, 2006, 2008 [23, 24]; Hjelm, 2012, 2018 [26, 27]; Kohinor, 2011a, 2011b [30, 31]; Mude, 2019, 2022 [38, 39]; Secker, 2002a, 2002b [47, 48]; Wallin, 2007, 2010 [52, 53]. Fig 1 presents the flow of studies through review and Table 1 presents a description of the selected studies. The 32 studies were conducted in 10 countries: Australia (n = 6), Canada (n = 1), Denmark (n = 1), France (n = 1), Netherlands (n = 4), Norway (n = 1), Sweden (n = 2), Switzerland (n = 1), United Kingdom (UK (n = 11), and the United States (US) (n = 4).

The studies focused on 8 conditions: hypertension (n = 3); diabetes (n = 9); sickle cell disease (n = 2), mental health conditions (n = 12); prostate cancer (n = 2); chronic hepatitis (n = 2); chronic pain (n = 1); and musculoskeletal disorders (n = 1).

One study focused on 'chronic conditions'. Participant health status was not directly sought in the study; however during recruitment three participants volunteered that they lived with hypertension, diabetes and asthma. Another study examined perspectives on mental health and mental illness, but one participant volunteered having had a mental health crisis in the past.

Five studies focused on experiences of co-morbid and multi-morbid conditions: participants lived with hepatitis B and liver cancer [16], hypertension and diabetes [42], diabetes and overweight [23]; diabetes, high cholesterol and high blood pressure [19]; obesity, hypertension, diabetes and heart attacks [25]. Two studies highlighted the mental health outcomes of living with chronic pain [36, 41]. Methodologically, 23 papers reported using individual interviews only, 2 used focus group discussions only, and 7 used mixed qualitative methods. The mixed method studies blended individual interviews and focus group discussions (N = 5), interviews and participant observation (n = 1) and interviews and 'case sampling' (N = 1).

Cumulatively, the studies focused on the perspectives of 748 individuals, from 27 African countries: Burundi, Cameroon, Chad, Congo, Democratic Republic of Congo, Egypt, Eritrea, Ethiopia, Ghana, Guinea, Ivory Coast, Liberia, Mali, Mauritania, Morocco, Nigeria, Kenya, Rwanda, Senegal, Sierra Leone, Somalia, South Sudan, Sudan, Tanzania, Uganda and Zimbabwe. Some studies stated participants' regions of origin, for example 'West Africa' [33].

The majority of studies described participants through a lens of immigration and applied three categories: 'immigrants' or 'migrants', 'asylum seekers' and 'refugees'. In the case of studies conducted with established immigrant communities in countries like the UK, US and Netherlands, formal terms of identification used in these countries for minoritized ethnic

**Table 1. Description of included studies.**

| Study (country) | Year of study | Condition(s) | Study type | Participants and settings | Description of population |
|---|---|---|---|---|---|
| Allard (2018) (Australia) [16] | 2018 | Chronic hepatitis B Liver cancer | Interviews | N: 19 Gender: 8m, 11w Age: 18–64 Location: Victoria Setting: hospital; 3 community clinics | "African-Australians" (Ethiopia, Sudan, South Sudan, Somalia) |
| Alloh (2019) (UK) [17] | 2017–2018 | T2D | Interviews | N: 34 Gender: 15m,19m Age: 33–82 Location: London Setting: Community, support groups; churches; mosques | "West African Immigrants (WAIs)" [Nigeria, Ghana, Gambia] |
| Alloh (2021) (UK) [18] | 2017–2018 | T2D | Interviews | N: 34 Gender: 15m,19m Age: 33–82 Location: London Setting: Community, support groups; churches; mosques | "West African Immigrants (WAIs)" [Nigeria, Ghana, Gambia] |
| Anderson (2013) (UK) [19] | NS | Prostate cancer | Interviews | N = 3 African men (Total– 7) Gender: men Age: 60–76 Location: South England Setting: Healthcare | "African and Afro-Caribbean (AAC)"men |
| Asgary (2011) (USA) [20] | 2007–2009 | Mental health | Interviews FGD | N: 35 Gender: 30m; 5w Age: under 40 Location: New York Setting: 2 clinics | "asylum seekers" from 12 named countries: Cameroon, Chad, Congo, Egypt, Eritrea, Ghana, Guinea, Ivory Coast, Mali, Mauritania, Senegal, Sierra Leone |
| Bamidele (2019) (UK) [21] | 2016–2018 | Prostate cancer | Interviews FGDs | N: 47 Gender: 25men Age: 65 yrs or less Location: England Setting: Mixed, home, work, HCF | Black African (BA) and Black Caribbean (BC) men |
| Bettemann (2015) (US) [22] | 2009–2010 | Mental illness | Interviews | N: 20 Gender: 10m. 10w Age: n/a Location: Salt lake City Setting: community | Somali, Somali Bantu |
| Beune (2006) (Netherlands) [23] | NS | Hypertension | Interviews | N: 19 (total 54) Gender: 20m, 26w Age: 35-65yrs Location: Amsterdam Setting: Primary care centres | African-Surinamese (hereafter, Surinamese) immigrants from the former Dutch colony of Suriname and Ghanaians from West Africa |
| Beune (2008) (Netherlands) [24] | NS | Hypertension | Interviews | N:19 (total 54) Gender: 20m, 26w Age: 35–65 Location: Amsterdam Setting: primary care centres | African-Surinamese; Surinamese; Ghanaians |
| Cooper (2012) (UK) [25] | NS | Chronic diseases | Interviews | N: 19 Gender:10m,19w Age: 18–60 Location: Glasgow Setting: community organisations, 2 protestant churches, 5 'African societies based on language or country of origin' | "French and Swahili speaking African migrants" from: "Eastern Africa (Somalia/Tanzania/ Uganda /Rwanda/Burundi); Congo; Democratic Republic of the Congo/Republic of Congo"; 'Francophone West Africa' |
| Hjelm (2012) (Sweden) [26] | NS | Gestational diabetes | Interviews | N: 10 (total 23) Gender: 10 w Age: 23–41 Location: Sweden Setting: clinic; homes | African migrants [Somalians, Ethiopians] |

*(Continued)*

**Table 1.** (Continued)

| Study (country) | Year of study | Condition(s) | Study type | Participants and settings | Description of population |
|---|---|---|---|---|---|
| Hjelm (2018) (Sweden) [27] | NS | Gestational diabetes | Interviews | N: 9 (total 23)<br>Gender: 9w<br>Age: 23–40<br>Location: Sweden<br>Setting: clinic | African migrants<br>[Somalians, Ethiopians] |
| Jager (2019) (Netherlands) [28] | NS | T2D | Interviews | N: *2 African women* (total 12)<br>Gender: 2w (4m,8w)<br>Age: 69, 87 (44–87)<br>Location: Arnhem and Nijmegen regions, Netherlands<br>Setting: dietetic practices | 'first generation migrant respondents from African, Asian or Latin American origin' (2 from Morocco) |
| Kindarara (2017) (USA) [29] | 2016 | T2D | Interviews | N: 10<br>Gender: 5m, 5w<br>Age: 44–76<br>Location: Sacra-mento county<br>Setting: churches | "Sub-Saharan African immigrants" 'least studied racial subgroup |
| Kohinor (2011a) (Netherlands) [30] | 2008 | T2D | Interviews | N: 16 African Surinamese (total 32)<br>Gender:12m, 20w<br>Age: 36–70<br>Location: Amsterdam<br>Setting: family general practices | 1st generation African Surinamese and Asian Surinamese<br>(West African heritage) |
| Kohinor (2011b) (Netherlands) [31] | 2008 | T2D | Interviews | N: 16 African Surinamese (total 32)<br>Gender: 12m, 20w<br>Age: 36–70<br>Location: Amsterdam<br>Setting: family general practices | 'African and Hindustani Surinamese' |
| Kokanovic (2008) (Australia) [32] | 2002–2003 | Depression Distress | Interviews FGDs | N: 62<br>Gender: NS<br>Age: NS<br>Location: Perth<br>Setting: community | East African refugees (Ethiopians, Somalis) |

| Study (country) | Year of study | Condition(s) | Study type | Population | Description of population |
|---|---|---|---|---|---|
| Maxwell (1999) (UK) [33] | NS | Sickle cell disease | Interviews Focus groups | N: *29 West Africans* (Total 57)<br>Gender: 25m, 32w<br>Age: 20–60<br>Location: London<br>Setting: hospital | 'Afro-Caribbean or West African' |
| Melamed (2019) (Switzerland) [34] | 2017 | Mental health | Interviews | N: 10<br>Gender: 10m<br>Age: 20–35<br>Location: Basel<br>Setting: Onsite at a refugee home and nonclinical setting | Eritrean asylum-seekers<br>Tigrinyan |
| Memon (2016) (UK) [35] | 2010 | Mental health | FGD | N: *6 Black/Black British* (total– 26)<br>Gender: 13m, 13w<br>Age: 18+<br>Location: Brighton Southeast England<br>Setting: community centre | Black and minority ethnic (BME) communities Asian/Asian<br>British; Black/Black British; mixed communities; 13 participants did not respond to the question on ethnicity. |
| Michaelis (2015) (Denmark) [36] | 2014 | Chronic pain | Interviews | N: *2 African women* (total 13)<br>Gender: 13w<br>Age: 50–60 (33–63)<br>Location: Copenhagen<br>Setting: OP services | 'immigrant women' originally from Pakistan, Iraq, Jordan, Turkey, Afghanistan, Somalia and Morocco |
| Michlig (2022) (US) [37] | Oct 2017-Nov 2018 | Mental health/ illness | FGDs | N = 168<br>Gender: m-84; w-84<br>Age: 14+<br>Location: Phoenix, Tucson.<br>Setting: community | Somali, Ethnic Somali, Somali Bantu, Somali Maay Maay. |

(*Continued*)

**Table 1.** (Continued)

| Study (country) | Year of study | Condition(s) | Study type | Participants and settings | Description of population |
|---|---|---|---|---|---|
| Mude (2019) (Australia) [38] | June 2015- May 2016 | Chronic hepatitis B | Interviews | N = 15<br>Gender: 8w; 7m<br>Age: 18–65<br>Location: Adelaide<br>Setting: community | South Sudanese born in South Sudan |
| Mude (2022) (Australia) [39] | June 2015- May 2016 | Chronic hepatitis B | Interviews | N = 15<br>Gender: 8w; 7m<br>Age: 18–65<br>Location: Adelaide<br>Setting: community | South Sudanese born in South Sudan |
| Noakes (2010) (UK) [40] | NS | T2D | FGD | N: *5 African women* (Total -13)<br>Gender: 5m, 8w<br>Age: 44–77<br>Location: London<br>Setting: South London NHS Hospital Trust | 'Black African/African-Caribbean adults with type 2 diabetes; English speaking' |
| Nortvedt (2016) (Norway) [41] | NS | Chronic pain | Interviews Participant observation | N: *4 from African countries* (Total—14)<br>Gender: 14w<br>Age: 30–56<br>Location: South Norway<br>Setting: Rehabilitation hospital; outpatient clinic | 'adult immigrant women from Asia and Africa'- 'three from North Africa, one from East Africa' |
| Nyaaba (2019) (Netherlands) [42] | Aug-Oct 2016 | Hypertension | Interviews | N: *20 based in Amsterdam* (Total- 55)<br>Gender: 14w, 41m<br>Age: 30–73<br>Location: Amsterdam, Kumasi, Tamale, Bolgatanga<br>Setting: community | Ghanaian migrants living in the Netherlands |
| Omar (2017) (Australia) [43] | 2013–2014 | Mental illness | FGDs Interviews | N: 36<br>Gender: 36m<br>Age: 18–60<br>Location: Melbourne<br>Setting: community | Horn of Africa Muslim men; Somalia, Somali Australians, Eritreans |
| Renedo (2019) (UK) [44] | 2016–2017 | Sickle cell disease | Interviews (longitudinal design) | N = 48*<br>Gender: 30w, 18m<br>Age: 13–21<br>Location: London; 'one other English city'<br>Setting: Communities, hospitals | "Black African and Afro-Caribbean groups" *numbers not disaggregated by African and African-Caribbean heritage. |
| Said (2021) (Australia) [45] | Dec 2017-March 2018 | Mental illness/ health | Interviews | N = 31<br>Gender: 31w<br>Age: 18–34<br>Location: Melbourne<br>Setting: community | 1st generation Somali-Australian (21 born in Australia; 10 migrated at early ages) |
| Sauvegrain (2017) (France) [46] | 2015–2016 | Hypertension Pre-eclampsia | Interviews | N: 33<br>Gender: 33w<br>Age: 22–45<br>Location: Paris<br>Setting: Three public maternity units | 'Immigrants from sub-Saharan Africa (SSA)' |
| Secker (2002a) (UK) [47] | NS | Mental illness | Interviews | N: *6 of African heritage* (Total– 26)<br>Gender: 10m, 16w<br>Age: 18–64<br>Location: London<br>Setting: Mental health resource centre | 'African and African-Caribbean' |

(*Continued*)

**Table 1.** (Continued)

| Study (country) | Year of study | Condition(s) | Study type | Participants and settings | Description of population |
|---|---|---|---|---|---|
| Secker (2002b) (UK) [48] | NS | Mental illness | Interviews | N: *6 of African heritage* (Total– 26)<br>Gender: 10m, 16w<br>Age:18–64<br>Location: London<br>Setting: Mental health resource centre | 'African and African-Caribbean' |
| Siad (2018) (Canada) [49] | 2015–2016 | Gestational diabetes | Interviews | N: 10<br>Gender: 10w<br>Age:18+<br>Location: Calgary<br>Setting: Diabetes-in-pregnancy (DIP) clinics | East African immigrant women; [Somali, Ethiopian] |
| Smith (2019) (Australia) [50] | NS | Mental health | Interviews FGDs | N: *2 from Africa*?* (Total—24)<br>Gender: m/w<br>Age: 'adults and youth'<br>Location: Launceston, Australia<br>Setting: community | Former refuges from Afghanistan Bhutan, Burma, Sierra Leone, Sudan and Iran (*all but 2 participants are linked to national background) |
| Wagstaff (2018) (UK) [51] | NS | Schizophrenia | Interviews | N = 7*<br>Gender: 7m<br>Age: 31–64<br>Location: West Midlands:<br>Setting: community | 'Black men' (* African heritage not stated |
| Wallin (2007) (Sweden) [52] | NS | T2D | Interviews | N: 19<br>Gender: 8m, 11w<br>Age: 30–83<br>Location: Sweden<br>Setting: Primary health centres | Immigrants from Somalia; refugees from Somalia |
| Wallin (2010) (Sweden) [53] | NS | T2D | Interviews | N: 19<br>Gender: 8m, 11w<br>Age: 30–83<br>Location: Sweden<br>Setting: Primary health centres, Public places and work place | 'immigrants from Somalia'; Somalis |
| Weich (2012) (UK) [54] | NS | Schizophrenia | Interviews Case sampling | N = *1 African man* (total– 40)<br>Gender: 22m, 18w<br>Age: 18–65<br>Location: Birmingham<br>Setting: PCT | "Black (n = 8), Black British = 3, Black-Caribbean = 3, Mixed = 1 Black African = 1 along with South Asian (n = 32) and White British (n = 32)" service users and carers recruited; 8 interviewed. |

groups were applied, such as 'African Surinamese' (e.g Buene et al, 2006, 2008 [23], [24], and 'Black British' (e.g. Memon, 2016 [35]). With the exception of two studies which described participants as "1st generation African Surinamese' [30], [31], and '1st generation Somali-Australian' [45] the latter set of studies did not analyse participants' generational status although generational differences were implied in the results and selected participant narratives.

Other forms of participant descriptors included linguistic groups or 'native language' of participants (e.g Cooper et al, 2012 [25]) and religious affiliation (e.g Omar et al, 2017 [43]).

In terms of language used in the studies, English language competence was an inclusion criterion for the majority of studies. A few studies, focused on refugee communities, recruited participants who communicated in their native languages and used interviewers/facilitators with linguistic competence.

Table 2 captures the nuances of participant descriptions, focusing on countries and immigration status of participants, for studies that provide this data. Full details are provided in Tables 1 and 3.

**Table 2. Description of participants.**

| Description of participants | No. | References |
|---|---|---|
| General regional label of 'Africans' or 'Black Africans' | 15 | [17–19, 21, 29, 33, 35, 40, 41, 44, 46–48, 51, 54] |
| National descriptors (e.g., Eritrean, Ghanaian, Nigerian, Somali, Surinamese) | 12 | [20, 22, 28, 34, 37–39, 42, 43, 50, 52, 53] |
| Combination of general label and national descriptors (e.g., "African-Australians" describing communities of Ethiopian, Sudanese, South Sudanese, Somali heritage) | 8 | [16, 23, 24, 26, 27, 32, 36, 49] |
| Linguistic groups or native language of participants (e.g., Somali Bantu) | 3 | [22, 25, 37] |
| Religious affiliation as descriptors (e.g., 'Horn of Africa Muslim men') | 1 | [43] |
| Generational descriptors (e.g. 1st generation Somali-Australians) | 3 | [30, 31, 45] |

**Table 3. Profile of participants' African nationalities or countries of heritage and focal conditions.**

| Country (No of studies) | African nationalities (*chronic conditions in focus*) |
|---|---|
| Australia (4) | Allard (2018) [16]: Ethiopia, South Sudan, Sudan, Somalia (*chronic hepatitis B*) |
| | Kokanovic (2008) [32]: Ethiopia, Somalia (*depression*) |
| | Mude (2019, 2022) [38, 39]: South Sudan (*chronic hepatitis B*) |
| | Smith (2019) [50]: Sierra Leone, Sudan (*mental health*) |
| Canada (1) | Siad (2018) [49]: Eritrea, Ethiopia, Somalia, South Sudan (*gestational diabetes*) |
| Denmark (1) | Michaelis (2015) [36]: Morocco, Somalia (*Chronic pain*) |
| France (1) | Sauvegrain (2017) [46]: Angola, Cameroon, Comoros, Congo, DRC, Ivory Coast, Nigeria, Senegal (*hypertension, pre-eclampsia*) |
| Netherlands (4) | Beune (2006, 2008) [23, 24]: Ghana, Surinam (*hypertension*) |
| | Jager (2019) [28]: Morocco (*type 2 diabetes*) |
| | Kohinor (2011a, 2011b) [30, 31]: Surinam (*type 2 diabetes*) |
| | Nyaaba (2019) [42]: Ghana (*hypertension*) |
| Norway (1) | Nortvedt (2016) [41]: 'North and East Africa' (*Chronic pain*) |
| Sweden (2) | Hjelm (2012, 2018) [26, 27]: Somalia, Ethiopians (*gestational diabetes*) |
| | Wallin (2007, 2010) [52, 53]: Somalia (*type 2 diabetes*) |
| Switzerland (1) | Melamed (2019): [19]: Eritrea (*mental health*) |
| UK (11) | Alloh (2019, 2021) [17, 18]: Gambia, Ghana, Nigeria (*type 2 diabetes*) |
| | Anderson (2013) [19]: "African men" (*prostate cancer*) |
| | Bamidele (2019) [21]: "Black African" (*prostate cancer*) |
| | Cooper (2012) [25]: Burundi, Congo, DR Congo, Kenya, Rwanda, Somalia, Tanzania, Uganda; Francophone W. Africa ("*chronic diseases*") |
| | Maxwell (1999) [33]: "West African" (*sickle cell disease*) |
| | Memon (2016) [35]: "Black/Black British" (*mental health*) |
| | Noakes (2010) [40]: "Black African" (*type 2 diabetes*) |
| | Renedo (2019): [44]: "Black African groups" (*sickle cell disease*) |
| | Secker (2002a, 2002b) [47, 48]: "African" (*mental illness*) |
| | Wagstaff (2018) [51]: "Black men" (*Schizophrenia*) |
| | Weich (2012) [54]: "Black African" (*Schizophrenia*) |
| US (4) | Asgary [20]: Cameroon, Chad, Congo, Egypt, Eritrea, Guinea, Ghana, Ivory Coast, Mali, Mauritania, Senegal, Sierra Leone (*mental health*) |
| | Bettemann (2015) [22]: Somalia (*mental illness*) |
| | Kindarara (2017) [29]: Eritrea, Ethiopia, Kenya, Liberia, Nigeria, Sierra Leone, Zimbabwe (*type 2 diabetes*) |
| | Michlig (2022) [37]: Somalia (*mental health*) |

### 3.1. Narrative synthesis

We synthesise the results in three sections: (1) how individuals make sense of chronic conditions; (2) experiences and impact of chronic conditions; (3) illness action.

For each section, we present dominant themes followed by minor themes, and outliers where these exist. We also highlight areas of consensus and conflict within and across studies. Quotes from studies that capture the essence of themes will be presented. At first mention of each study, we provide a full description offered by the original authors, followed by a short description. For example "in a study conducted among French and Swahili-speaking African immigrant community from East Africa (Somalia, Tanzania, Uganda, Rwanda and Burundi and the Congo/Republic of Congo in the UK [25] (hereafter East African immigrants in the UK)". For subsequent mentions we use the short description, for example, "in the study with East African immigrants in the UK".

**3.1.1 How individuals living with chronic conditions make sense of chronic conditions.**   Fifteen studies conducted in six countries (Australia, Netherlands, Switzerland, Sweden, UK, US) explored understandings of chronic conditions [19, 22–27, 29–32, 34, 35, 42, 43] Two studies–conducted among French and Swahili-speaking African immigrants from East Africa (Somalia, Tanzania, Uganda, Rwanda and Burundi and the Congo/Republic of Congo in the UK [25] (hereafter East African immigrants in the UK) and African-Surinamese living with Type 2 diabetes in the Netherlands [31] (hereafter African-Surinamese in the Netherlands)—reported lay definitions of chronic illness. In both studies participants defined chronic illnesses as 'permanent' or incurable' diseases.

*'a chronic disease is a disease which keeps coming back..'* (17M 40–49 French, in Cooper et al, 2012 [25]).

In the majority of studies, lay understandings of chronic conditions emerged when participants discussed the causes of specific conditions such as hypertension, diabetes, strokes, cancers, obesity and mental health conditions. Participants from six countries (Netherlands, United Kingdom, United States, Australia, Switzerland and Sweden) attributed chronic illness to multi-level factors:

1. *family histories* (among Ghanaians and African-Surinamese in the Netherlands living with hypertension [23], East African immigrants living in the UK [25], immigrant African women with gestational diabetes living in Sweden [26] and Somali and Ethiopian immigrants in the US living with Type 2 diabetes [29];

2. *individual lifestyles* in particular eating unhealthy foods high in salt, fat and spices, physical inactivity and obesity) (African Surinamese in the Netherlands living with hypertension and diabetes [24, 30], Ghanaians in the Netherlands living with hypertension [24]; East African immigrants in the UK [25], and 'sub-Saharan African immigrants from Eritrea, Ethiopia, Kenya, Liberia, Nigeria, Sierra Leone and Zimbabwe' living with Type 2 diabetes in the United States [29] (hereafter East, West and Southern African immigrants in the US);

3. *psychosocial stressors* (Ghanaians and African-Surinamese living with hypertension in the Netherlands [23, 42], Somali and Ethiopian women with gestational diabetes living in Sweden [26], East, West and Southern African immigrants living with Type 2 diabetes in the United States [29], Eritrean Asylum seekers in Switzerland [34], and 'Horn of Africa Muslim men from Somalia and Eritrea in Australia [43] (hereafter Somali and Eritrean men in Australia);

4. *structural factors* (e.g, the stress of work, toxic foods) among East African immigrants in the UK [25], Ghanaian and African-Surinamese in the Netherlands [24]; and

5. *supernatural factors* among a Somali refugee community in the US [22], West African communities in the UK [18]

*We have people who are naturally BP and we have people who create BP.am a natural BP because my father had BP (Ghanaian female: ID5) [23]*

*I can say that 90% of the people, my home people who came from Africa, they never have diabetes when they were in my country but when they come here [US], they have it. So I think the lifestyle, the competition, the rent, everything affects their life. Sometimes even if you are working, sometimes you cannot afford to pay the rent. It is expensive. So it adds up to the stress... then stress is another cause. (P5, Kindarara, 2017 [29])*

*"But you know our culture, it was thought to be something else, someone has bewitched me so I concluded that something was wrong"* (Orisa 52 years, male diagnosed in West Africa, Alloh 2021 [18])

*"There are some people who get possessed, we believe in that. Those people that were crazy, you read Qur'an on them, you do a lot of imam talk, and you pray for them."* [22]

In five studies conducted in the UK [19, 25, 35]; Sweden [26] and United States [29] participants living with gestational diabetes, prostate cancer and depression, stated they were not aware of some conditions prior to their own diagnosis, or the diagnosis of significant others.

*. if we could get information about gestational diabetes, how you can prevent it, when you can get symptoms and which. . .then I could have been rescued earlier. . .if I knew about preventive measures. . .I would follow them. . . (A) (Hjelm, 2012 [26]) (Woman in Hjelm, 2012 [26])*

*The elder brother of one of my friends died suddenly, and then he said he died of prostate, and I said, "What's prostate"? And he decided to tell me about it. . . And I was ashamed that I've not been tested myself, because I was close to 60 then'* [29]

Three studies conducted in the Netherlands [23, 42] and UK [17] reported that some participants living with hypertension and diabetes believed, at the early stages of their diagnosis, that their condition could be cured. This perspective changed over time.

*Oh, I didn't understand it will take me to this time [chronic]. I thought it will go [cure] so I took medicine for some time... I used the herbal medicine in the beginning but it [HTN] didn't all go... it [BP] was always up, the nurse advised me so now, I don't joke with my medicine."* *(IDI-Male-11 in Nyaaba, 2019 [42])*

*I was in a desperate state at this point and was willing to do anything to cure my diabetes. I drank my own urine, drank ewuro (ewuro (Vernonia amygdalina) is also known as bitter leaf), ate onions and garlic but no cure".* *(Yeriyaya 77 years, male in Alloh, 2019 [17])*

In the study conducted among African-Surinamese living with Type 2 diabetes in the Netherlands, one participant did not that view diabetes as chronic but rather a temporary illness [31]

*I don't believe I have diabetes. It's just there for a little while (70 year-old African Surinamese woman in Kohinor 2011a [31])*

Across the studies, participants drew on eclectic sources for information on chronic diseases, including: family members, mass media (radio and television) and healthcare providers (doctors, nurses). In one study among Ghanaians in the Netherlands [42], participants sourced health information from the Dutch and Ghanaian mass media.

*You know we listen to radio on the [inter]net...we hear about the medicines from the Ghana FMs... oh, you family member can buy and send to you if you send the money (IDI-Male-01in Nyaaba, 2019* [42])

*3.1.1.1 How specific chronic conditions are understood.* Three studies examined definitions and understanding of specific chronic conditions: hypertension, diabetes, stroke, cancers and mental illness. Among Ghanaians in the Netherlands and East Africans in the UK, hypertension was commonly defined as 'ill heart' or heart under strain [23, 25]. Ghanaian participants also defined hypertension as having 'excess blood'[23]. East African participants in the UK understood hypertension as having 'high pressure' which was distinct from 'hypotension' or 'low pressure', and strokes as problems with nerves in the head [25].

*you can't sweat, so everything stays in your body: Too much blood! But if you are in Africa, you sweat (45-year-old Ghanaian female: ID5) in Beune, 2006* [23])

*hypertension it is when you have, your pressure is very high and hypotension is when your pressure is very, too low' (13M 30–39 French) in Cooper, 2012* [25])

*stroke 'it is like... a crack in the nerves of the head, I think' (17M 40–49 French, in Cooper et al,2012* [25]*).*

Diabetes was defined as 'having excess sugar' by East African participants in the UK [25] and West, East and Southern African participants in the US [29]. Participants in both studies understood diabetes through the body's use of insulin. While UK participants explained diabetes as 'balancing insulin level', US participants explained diabetes in terms of the body's inability to produce enough insulin or lack of it in the body'.

*They say perhaps [you get diabetes] if you are eating a lot of sugar. I argue with my young boy here. He loves sugar. When he drinks tea he just puts in a lot of sugar. Me I [do] tell him. Perhaps the thing that is helping him is those [physical] exercises. (11F 40–49 Swahili) in Cooper, 2012* [25])

*it's like it's your sugar level and like constantly you have to inject yourself with insulin like trying to balance' (10M 18–24 English). (Cooper, 2012* [25])

*Diabetes is a disease whereby your body is not able to give you enough insulin to cover the glucose or the sugar that's in there that came from food and other sources." (P3)* [29]

Understandings of chronic mental illness were explored in five studies in four countries: among Somali and Ethiopian refugee men in Australia [32, 43], Eritrean Asylum-seekers in Switzerland [34] and Somali refugees in the US [22]. These studies reported two main categories of mental illness; less severe or common mental illness (typically depression) and more serious form of mental illness (typically psychosis). Immigrant men and women in Australia understood depression as a continuum of personal and social problems, from general worry about life circumstances to the stress of social isolation or poor integration post migration. Ideas about mental health problems were expressed through cultural lens, with comparisons

made between how these terms were understood in participants' home countries versus Australia.

*"We say like that person is worried, it is a worry thing inside the head. It is hard for us to find words like the depression and all those kind of mental language. It is a sort of like that person is a worry, like when they get a lot of worry that just happens to them. (Ayan, Somali woman, in Kokanovic et al, 2008* [32]*)*

*your social [well being] and health depend on our settlement situation, I think it's clear that if you are not resettled well you cannot have social and mental health. (Dalmar, Somali man, in Kokanovic et al, 2008* [32]*)*

*[depression] is not understood the way that it is looked at in such a specific manner in the Western culture and it is not that much experiencedy because if someone feels a little bit down there is always someone around. (Azmera, Ethiopian man, in Kokanovic et al, 2008* [32]*)*

Severe forms of mental illness were described with terms such as 'crazy', 'madness' and 'off-minded' and associated with loss of control in this group as well as among the Eritrean group in Switzerland [34].

*"unless it's the worst phase, that he's going off-minded, is completely sick, like psychiatric symptoms,.. . small mental health related symptoms like stress, they are usually not considered as a sickness, so priority is given to the physical sickness" (male, age 30–35 years, in Melamed, 2019* [34]*)*

*[It] is madness in which the person jumps around and shouts without reason. He sometimes throws his clothes and walks naked* [43]

The Somali refugee group in the United States observed that individuals with severe forms of mental illness were more harmful and violent compared to those with less severe illness [22]. Some participants offered nuanced descriptions of severe mental illness, distinguishing between quiet and violent versions.

*[In] the Somali context, [it] is someone who has totally lost it, violent, naked [and] should be on chains. . .* [43]

*There are different kinds of crazy. There are some who doesn't do anything, just sitting down, doesn't talk to anyone. He [is] just by himself. And there is one who hit people, hurts people, they run around."* [22]

Participants in five countries—Netherlands, Sweden, Australia, US and Switzerland—attributed causes of (specific) chronic physical and mental illness to psychosocial/emotional stress. Psychosocial stress was perceived as a major cause of hypertension among Ghanaians and African-Surinamese with hypertension in the Netherlands [23, 42], Somali and Ethiopian women with gestational diabetes in Sweden [26]; of diabetes among West, East and Southern African immigrants in the US [29]; and of mental illness among Eritreans in Switzerland [34], Somali immigrants in the US [22] and Somali, Eritrean and Ethiopian communities in Australia [32, 43].

*there are also mental health problems for these people who have stress, and then they are isolated, they are away from themselves, so this is what I see... I see some peo- ple like they take the marijuana, cocaine" (male, age 20–24 years, in Melamed, 2019* [34]*)*

In the Netherlands, Ghanaian and African-Surinamese participants understood psychosocial/emotional stress mainly in terms of the financial pressure they faced in sustaining their livelihood (such as paying rent and bills) and pressure from family back home [23]. For Eritreans in Switzerland, the need to provide for family 'back home' led to engagement in multiple jobs which put both their physical and mental health at risk.

> *I was under pressure because four families from my extended family who were living in Nairobi were dependent on me. . . I was a taxi driver for five days a week and working at factory for five night shifts. . . . Because of that huge pressure, you will finally reach to a level you can't sleep during night. You will be affected with a lot of stresses, frustration and depression that will indirectly destroy your health.(Elderly man in Omar, 2017 [43])*

Prejudice emerged as a source of psychosocial stress in studies in the Netherlands, UK and Australia. In the Netherlands, an African-Surinamese participant spoke of negative experiences with police [23]; in the UK a participant associated the origin of his mental health problems with the experience of being black in a racist society in the UK (Secker, 2002a); and in Australia Somali and Ethiopian participants attributed poor mental health to discrimination and racism [32].

> *Everybody who gets high BP is stressed, stressed! When you open your mailbox, bills, bills, only bills, for months on end! Yeah, as Surinamese, we are not used to such things, eh! When you come out of the bar, immediately the police is on your case. Pull you over, ask for ID (58-year-old Surinamese male: ID31 in Beune, 2006 [23])*

> *I found that the staff didn't fully understand, and this is because of my colour and my upbringing, didn't fully understand my concerns about my own well-being and the problems of being black and still being in the mental health syste*m [47].

Three studies—two in Australia, one in the US—examined gender differences in mental illness attributions. In the US study with Somali immigrants [22]—men attributed mental illness to social status stressors, in particular men's inability to marry the women they loved due to poverty and unemployment. Somali and Eritrean men in Australia reported that men experienced feelings of worthlessness and emotional instability due to a loss of authority over women–this loss was linked to changing social conditions post-migration [43]. Women in one Australian study [32], emphasised family separation as the most pervasive source of emotional distress.

> *"the whole role changes you know, because there are role changes for women here and men, some men would feel they have been disempowered, disempowered because of new situation" (Iman, Somali man, in Kokanovic et al, 2008 [32]).*

> *'Al-Qawaamah' leadership and authorities, which men had over their families back home have been lost. His power [an African man] is challenged . . . He comes out of his house and you can see on his face huge frustration caused by bickering and yelling happening at home . . . (Older man in Omar, 2017 [43]).*

> *"Well, about how to overcome missing family, for example you come here, for example, I came here myself and my children are well fed but we do not havey my kids do not have a father, an auntie, for example their father is back home and they don't know what could possibly happen to him." (Zahara, Ethiopian woman, in Kokanovic, 2008 [32])*

**3.1.2 Experiences of chronic conditions.** Twenty-three studies conducted in nine countries (Australia, Canada, Denmark, France, Netherlands, Norway, Switzerland, UK and USA) focused on experiences of seven conditions (cancers, mental health conditions, chronic pain, diabetes, chronic hepatitis B, hypertension and sickle cell disease) [16, 19–24, 26, 29, 32–36, 40, 41, 44, 46–48, 51, 54, 55]. The studies reported the negative impact of chronic conditions predominantly and in the form of physical, psycho-emotional, social and financial disruptions.

*3.1.2.1 Physical disruptions*. Physical disruptions emerged as a dominant impact of chronic conditions. Two main forms were identified: bodily disruptions [16, 20, 29, 47], and sexual dysfunction [19–21, 24]. Individuals living with diabetes and hypertension experienced physical disruptions, ranging from mild bodily symptoms such as headaches, leg pains, dizziness, swelling sensations, and swollen feet to severe problems and complications including sores, eye problems, and organ damage, with reference to the liver and kidney.

> *You may be losing that [your eyes].. . because of the spoilage of the nerve endings in your eye.... Your extremities may start feeling some sensation, losing a sensation because of your nerve endings getting affected by the sugar that you have. Sometimes may be in very extreme cases a person may develop sores or wounds that are due to diabetes (P1)* [29].

> *When my BP is too high I see stars, or I become dizzy. Then I know that well: there is something wrong here. Then I go to the doctor in a hurry, like a bullet! Yes, my body gives clear indications, really! (45-year-old Surinamese female: ID22)* [23].

In the UK study on type 2 diabetes experiences, women reported struggling with their domestic chores when they went into hypoglycaemia:

> *I think it's rather dangerous for a diabetic to be on his or her own you know, because I find that when I go into hypoglycaemic, sometimes it's a struggle to maybe get to the kitchen."* [40]

Sexual dysfunction emerged as a major problem for men. Four studies conducted in the Netherlands, UK and US [19–21, 24] reported the concerns and anxieties expressed by male participants living with diabetes and prostate cancer about erectile dysfunction and impotence.

> *"I mean this thing has really hit me . . . previously we could just get on with it (sex) but now I can't, and even the fact that I have to go and inject myself before I have sex, that is my headache . . . and sometimes injecting yourself and it is not firm enough to have sex makes it even worse, ok I am not bothered, I won't do it, but I have to remember that I am not single, I am in a relationship and sex is a big part of the relationship, so if I am being selfish with it and didn't think about the impact on my wife, I'm not being honest with myself and it is not nice, so whether I like it or not I just have to . . ." (56 year old man in Asgary, 2011* [20])

*3.1.2.2 Psycho-emotional disruption*. Studies conducted in Denmark [36] (with immigrant women from seven countries including Somalia and Morocco living with chronic pain–hereafter Somali and Moroccan women with chronic pain), Norway [41] (with African immigrant women living with chronic pain), the UK (with men living with prostate cancer [19, 21], the Netherlands (with Surinamese and Ghanaians living with hypertension [23], the US [20] (with asylum seekers from 12 West, East and Central African countries) and Australia (with Ethiopian and Somali refugees) [32] reported that living with chronic pain, prostate cancer, hypertension, and mental health conditions, negatively impacted the psychological wellbeing of participants. Participants reported experiencing hopelessness and depression:

*"There is no future for us. I don't feel like I will be the same" [FG]* [20].

*The doctor does not understand what is wrong with me. They only understand that I have depression and because of the depression I have this pain. I tell them: "No, I have not got depression, it is because of the pain I am depressed—because I am always in pai*n [36].

*I have been depressed, I was sad and it was not ok. I didn't want to meet people; I wanted to be alone, just lying under the covers. This is how the days went by for a year (W5-I)* [41].

Participants reported living with fear and worry on a daily basis. The fear manifested at the personal and interpersonal levels. A Surinamese woman expressed a chronic fear of dying:

*R: It is always at the back of my mind. Like if I worry, it goes up agai*n.

*I: What are you afraid of?*

*R: Of dying (laughs). . .sorry. That you. . .uh, maybe you just drop dead (65-year-old Surinamese female: ID25)* [23]

In the same study, a young Ghanaian man recounted his fear of dying in the Netherlands, where he did not have a large family network. He struggled with the tension between staying in the Netherlands in order to provide adequate education for his children and relocating to Ghana where he could die surrounded by his family:

*I have two children in Ghana, but I have two here of 6 and $4^{1/2}$ · · · When I die in Ghana I know my father and mother is there. My family is big there. My only problem is just my pressure. Sometimes, when it's gone, I like to stay here. To let my children, the ones born here, get the education here. But the moment it's coming and I start getting worried, I like to go home' (35-year-old Ghanaian male: ID35)* [23].

At the interpersonal level, emotional disruptions manifested as worry and anxieties about the impact of chronic illness experiences on spousal relationships. In the UK studies with men with prostate cancer, participants expressed constant fear and worry about emotional disconnection and marital separation.

*'Also, when the chemo[therapy] was really having its effect on me, I was really afraid that she [wife] would leave me'* [19].

Among Ethiopian and Somali participants in Australia [32] the double burden of mental health problems and the socio-cultural dislocation associated with immigration created problems in marital relationships.

*"what people expected in Australia and what they really faced even in terms of finance. There are many people who have had businesses and heaps of money. And for these people to come here and be on the dole and just sit and talk to the kids and you know that affects marital relationships." (Tamirat, Ethiopian man)*

*3.1.2.3 Social disruption.* The importance of social support was emphasized in several studies: participants drew on support from local care networks of family, friends and community groups (such as patient support and faith-based groups).

However, chronic conditions also impacted on social participation, identity management and social stigma. In term of social participation, two studies from UK [40] (with "Black African adults" living with diabetes) and Denmark [36] (with Somali and Moroccan women living with chronic pain) reported participants experience of social isolation.

In the Danish study [36], the women described feeling lonely and isolated, and both states interfered with their ability to care for their children or do house chores:

*I cannot cook or anything in the home or be responsible for my children as before* [36]

In the UK study with 'Black African men' living with prostate cancer [21], the participants described the ways in which cancer disrupted their masculine identities. The participants reported feeling emasculated and rendered infertile by cancer and cancer treatment. This had a knock on effect on their marital relationships.

*but er, you know with we Africans when they say you are not a man, it means ern, one thing that comes to mind is you can't have children, you see when you're not a man, you can't have children . . ..in my case even though I could still have activity, you know when they remove the prostate, that's it, you know, you don't have any more semen, you see. . . I feel bad about it, but at least I'm alive you know. . ." (Mr Ellis, BA, 74)* [21]

Deacon et al (2005, p. 85) [56] define disease stigma as 'negative social "baggage" associated with a disease that. . . is not justified by the medical effects of disease on the human body'. This definition fixes attention on the physical and social consequences of disease. In the Dutch study with Surinamese and Ghanaians living with hypertension, participants lived with anticipation and fear of 'disease stigma' within their communities and therefore did not want others to know about their condition [23]. In one US-based study on mental health with Somali communities, a similar theme emerged [37].

*Our people they are different. If you tell them your problem, they're going to talk. Then everybody knows. They think maybe you have a problem in your house. Or maybe your husband have another woman (39-year-old Ghanaian female: ID14)* [23].

*"I once suffered from depression [participant notes that at the time he was facing deportation and a marital separation] . . . Life got very difficult for me, hope fading. If I went to another Somali, I know they would have stigmatized me. I went for counseling where the doctor lent me an ear and gave me advice. I wasn't given any medication, rather he listened to me and gave me advice. I should have received the same from my community."* (Elderly Bantu man, Michlig et al, 2021)

Two studies exploring mental illness perspectives and experiences among West, East and Central African asylum seekers in the US [20] and 'Black men' in the UK (Wagstaff, 2018) reported fear of stigmatization and of 'courtesy stigma' (Goffman, 1963/1990) [57]. Participants living with schizophrenia in the UK feared losing trust and respect for their families [35]. Similar dynamics of managing shame and family identity within the context of mental illness were reported by young Somali-Australian women who felt caught between more open attitudes of illness disclosure within their generation (millennial and Gen Z), and the restrictions of cultural traditions held by older family members (mothers, fathers, uncles and aunts) [45].

*in our country if you hear about somebody in this family sick, is mental, you lose trust in this person and whole family, maybe relative as well. And, [. . .] they whisper about the whole family and maybe talk about grandmother, grandfather, and they lose respect"* [35].

*Mental health is deeply stigmatized. Most countries outside the U.S. don't see mental illness as we see it. PTSD is very common, but there is a great deal of stigma as well as lack of appreciation about treatment."* [20]

*"I feel comfortable talking about our mental health issues, but my mum will not. I have a relative who has a mental illness, but there is no forum to discuss this, at all. My mother says, 'Why are you throwing our ceeb (shame) out there'? (into the community)? And that's the thing I don't want (to bring shame to my family). Therefore, I think it is because of ceeb (shame) that we got to hide our (mental illness status). [FGD 1, Participant 3, Said 2021* [45]*)*

*3.1.2.4 Financial disruptions.* The financial impact of chronic illness emerged in terms of the financial burden of treatment and inability to work. The Danish [36] and Norwegian [41] studies with women living with chronic pain reported that chronic pain disrupted economic activities and financial circumstances.

*Firstly, I cannot work because of my pain, it affects my financial circumstances, and it affects me socially* [36].

For Somali and Ethiopian immigrant women in Canada living with gestational diabetes [49] and East, West and Southern African immigrants in the USA living with Type 2 diabetes [29], financial insecurity was associated with expensive treatment, and in particular diet management.

*Well, when you go to the stores you see that the healthy and good food is very expensive that only rich people could have. But the poor people go mostly to fast food. That is the only thing they can afford. (P5)* [29]

*They told me there is a market by [the] University of Calgary. There is bread there for people who have diabetes.... She told me $6 for one bread. . .. When they told me it is expensive, I didn't put in my mind to go there because of my house situation, financial situation. When you go somewhere, you need to know your pocket, what you have"* [49]

A strong link between financial and social disruptions was made across several participant narratives. Women living with chronic pain in Denmark and Norway, for example, linked work to social wellbeing, and therefore, their inability to work negatively impacted on their social lives: A Somali woman in Denmark living with chronic pain in her shoulders, arms and legs remarked:

*Firstly, I cannot work because of my pain, it affects my financial circumstances, and it affects me socially. It is like a line, you need to have all in succession, when you have a good economic situation, then you will also have it well socially. When you are healthy, then you are socially well too; it is a circle—when you are losing a ring, then you are losing it all. In my situation, many rings are missing, both money- and health-wise. Then you just feel empty inside* [36]

**3.1.3 What individuals do about their chronic conditions: Illness practices.**   Four sets of illness practices were reported: (1) biomedical management; (2) faith-based action; and (3)

traditional medical treatment, including care practices drawn from indigenous models and offered at home or through social networks. A fourth cross-cutting practice emerged across some studies–this was described by Omar et al (2017) [43] as 'therapeutic journeys': the navigation of treatment seeking across transnational spaces. Treatment seeking in this context encompassed biomedical, traditional medicine and faith healing.

*3.1.3.1 Biomedical management.* Biomedical management included taking medications, managing conditions through diet and physical activity, and engaging with healthcare services.

East, West and Southern Africans living with diabetes in the US [29], Ghanaian immigrants with hypertension in the Netherlands [42] and Africans with severe mental illness (schizophrenia) in the UK [54] stressed the importance of medical adherence.

*I know when I get up in the morning... one of the first things I do is check my sugar and take my pills.... I know if I don't take my pills, [or] If I don't check my sugar, I'll be in danger.... So it's just like something you don't forget. (Participant with diabetes in Kindarara, 2017 [29])*

*It's [medication] always in my bathroom... so I take it after I have brushed my teeth... it makes me weak so I take it only in the morning so that my body can be strong in the evening (laughing)" IDI- Male-09 in Nyaaba, 2019 [42])*

Ghanaians and African-Surinamese with hypertension in the Netherlands stated that they adhered to prescribed medications because they trusted their doctors, they believed in the efficacy of anti-hypertensive drugs in controlling high blood pressure and a lack of side effects [24]. Participants also discussed barriers to medical adherence, including financial difficulties, high cost of medicines and perceptions about the diminishing efficacy of medicines over time and family interference.

*Diet management.* For participants in Sweden [26, 52] Netherlands [42], Canada [49] and the UK [17] living with gestational diabetes, type 2 diabetes and hypertension, dietary management involved healthy eating, in particular consuming more vegetables and reducing salt and sugar intake.

*We don't use salt... My husband and mother have it many years before my own, so I am already used to cooking without salt, so we always eat without salt." IDI- Female -07 in Nyaaba, 2019 [42])*

*I don't eat as much sugar as before, I buy bread without sugar. (Woman in Hjelm, 2012 [26])*

However, participants in these studies faced barriers to healthy eating. The barriers included psychological (e.g. tastelessness of healthy foods, lack of motivation to change dietary practices), structural (high cost of healthy alternatives, lack of direction from dieticians) and cultural (attempting new foods and diets from different cultures, and the challenges with communal eating).

*"My wife will not cook this rich food for you... I add some magi sauce because I cannot eat... sometimes, I hide and add small salt because all die be die... she always hides the salt (laughing).. . she has it [HTN] too so now the children are not here, she cooks the way the doctor says." (IDI-Male-08 in Nyaaba et al, 2019 [42])*

*There are times when there is a party or gathering.... In my community or people from my country we get together a lot. It's hard because you get together with people you know around food. You say ok it's a gathering so I can eat more. You don't get anybody discouraging you.... The women I know, the older women in our community, they have diabetes. Nobody is*

*educating others. Saying hey, watch what you're eating... so it makes it very hard when you're gathering around the food to be able to manage your diabete*s. *(P3 in Kindarara, 2017* [29]*)*

*Lifestyle changes.* Participants in the same studies also reported engaging in broader lifestyle changes including taking up more physical activity and reducing alcohol consumption and smoking. Some participants incorporated lifestyle change into daily activities, such as getting off the bus a few stops early. Barriers to these changes included the psychological challenge of changing lifelong habits, time constraints (particularly due to pressures of work) and negative cultural attitudes (for example around habits perceived as contrary to cultural norms).

*For instance, he usually does like this when he comes to the town centre by bus*: *he gets off at [X] and then walks the whole distance to the mosque and then walks back to [X] and gets the bus home. He tries to do some walking every day". (Interviewee 19 Wallin, 2007* [53])

*They see you riding a bicycle and think, "Oh why is this big person riding bicycle*?*".. . you feel shy because we do not do that in Ghana. . . If you have the money, you can go and pay and exercise but we came look for money, so the work is too much no time." (IDI- Female -20 in Nyaaba, 2019* [42]*)*

*Experiences in healthcare settings.* Two studies reported factors that facilitated ideal care in healthcare settings in the UK and Australia. Factors included access to well-resourced mental health facilities which provided holistic mental health services, demonstration of care and love by mental health professionals, and easy access to community-based mental health support in the UK [47, 48, 54], and access to well-trained culturally competent mental professionals in Australia [32];

*When you have had mental health problems, well I find myself that you are isolated at home. Most of your friends . . . they seem to think less of you because you're not working and you've had mental health problems. So you're sort of isolated. Having the centre is a place where you can come and meet other people who have had problems with mental health and also to asso-ciate with other people who don't think less of you because of your past situation (participant in Secker 2002b* [48]*)*

*The best way [to deal with mental health issues] is to go through bi-cultural community edu-cators who have been trained. (Asad, Somali man in Kokanovic, 2008* [32]*)*

A larger number of studies, including the discussed, also highlighted barriers to ideal care in healthcare settings including: a lack of empathic care, biases in treatment, high cost of care, lack of access to healthcare professionals, language barriers and immigration problems. For a number of studies, these factors led to general sense of mistrust of health services.

In one of the oldest studies reviewed, participants with sickle cell disease in the UK reported they experienced stigma from hospital staff, over-regulation of their medication regime by nurses and feelings of neglect [33].

*You do tend to find certain nurses who like to overstep their bounds, they feel they know the best regime for your painkillers. . ." (Focus group participant in Maxwell, 1999)*

*"[The nurses] just seem to concentrate on the pethidine injections and that's it. I've been in days without having any assistance with my hygiene and personal care, and changing of the sheets and helping me with fluids—just basic stuff like that." (Interview 11 in Maxwell, 1999* [33]*)*

In another UK study on sickle cell disease experiences conducted two decades later, a group of young people reported a lack of sympathy and condescending attitudes from health professionals [44].

*"I sort of got ignored [in the ward] when I was calm and everything was fine, and I needed my tablets to be a constant, like every four hours. [. . .] I'd be left for about eight hours, and then suddenly I'd get like a really bad sickle attack [. . .] And I was even more stressed out because no one was listening, and I'd go for all these hours without like any tablets [pain relief]. And then the doctor would react after or during, whilst I was having a really bad sickle attack, or I'd be like crying, or in the bed like hunched over. And that was the only time that she would listen, but when I was calm and OK, or just sat there and go OK, I need my tablets, right, I need it now, I've been tracking the times, and they'd be like OK, and then not come back. [. . .] She'd only see when I was in pain, and then she'd react, and then it was too late. [. . .] It went on for about five days [. . .] the second I say I'm not fine, for doctors they don't seem to always react" (Z2 19–21 years old, in Renedo et al, 2019 [44]*

For Eritrean asylum-seekers in Switzerland and African participants in the UK [34, 47], and Somali-Australian women in Australia [45], delays and a lack of access to mental health professionals were reported as major barriers.

*The [manager] is the one who decides when you will see a doctor. So, he can make you still for a long... he can make you wait. It's him who decides what happens. It's not that you feel sick and you see [a doctor]. . . but you have to present to him and then he decides. If he thinks that tablets are enough for you, then he just gives you tablets and you go back. And then if you need more, after several returns, several recourses, he might be ready to [let you] see a doctor... it is not appropriate... what he decided to do, just to give tablets from the cupboard in his office, it's not appropriate. He should send people immediately [to a doctor]. (male, age 30–35 years, in Melamed, 2019 [34])*

Three studies conducted among former refugees from Sierra Leone and Sudan in Australia [50] and African participants in the UK [35, 54] reported that participants inability to communicate in English was a major barrier in seeking mental healthcare.

*"I know one person, he has struggled and struggled. He said what he could say but the limit of the words, the real words he wanted to say to the GP, he did not know the meaning of them in English. Then in the end they end up getting the wrong treatment" (Male B. FG2 in Memon, 2016 [35])*

Inadequate consultation time at health facilities was a barrier to care for participants seeking mental healthcare in the UK [35], Ghanaians seeking hypertension care in the Netherlands [42], and South Sudanese individuals seeking chronic hepatitis B care in Australia [39].

*You treat every- body with the same brush [. . .] you go in and [. . .] you are just like the next person. They do not take time to listen to you. And I do not think it is because of colour or anything like that. In talking to a lot of people, including White people, they encounter the same thing. . ." (Female F. FG1 in Memon, 2016 [35])*

*You know the doctors here; 10minutes. . .there is no time to ask questions... I speak the Dutch small, so it is small small... sometimes I don't hear anything... oh, people talk, you know?... ." IDI- Male-15 in Nyaaba, 2019 [42])*

*"I was told to see a doctor once a year. So, this year, I didn't attend my appointment because I was so angry. Why waste my time when they are not there to help me? They only give you like a few minutes, and that is it." (Mariam, 26-year-old female, in [Mude et al, 2022 [39]])*

In the study with East, West and Central African asylum seekers in the US, immigration problems (in particular lack of documentation) and the high cost of mental health treatment emerged as major barriers [20].

*"Many people are afraid to seek care because they might get arrested or reported. Only in emergency"* [20].

'Stigma consciousness' is the expectation of being stigmatized: this socio-psychological process can lead to internalization of anticipated stigma or projection onto others [58].

*"our people cannot go to mental health services because [. . .] he thinks it is only crazy people going there" (Male L. FG2 in Memon, 2016 [35])*

In one study on mental health in the UK, stigma consciousness informed decisions not to access mental health services. In another study on mental health in Somali communities in the US, stigma consciousness shaped the decision to seek medical care.

*"I once suffered from depression. . .. If I went to another Somali, I know they would have stigmatized me. I went for counseling where the doctor lent me an ear and gave me advice. I wasn't given any medication, rather he listened to me and gave me advice. I should have received the same from my community."* (Elderly Bantu man, Michlig et al, 2021)

The latter suggested a careful strategy of engagement in formal healthcare services that minimised the risk of disease stigma.

*3.1.3.2 Faith-based action.* Six studies conducted in the US [22, 29], Sweden [53], Switzerland [34] and Australia [32, 43] reported different ways participants used their faith to help them treat their condition. First, dependence/belief in God's power and grace—among East, West and Southern African immigrants in the United States [29] and Eritrean asylum-seekers in Switzerland [34]. Second, praying and reading the Qur'an for spiritual guidance for mental health problems such as depression and severe mental illness—among Somali refugees in the US [22] and Somali and Ethiopian refugees, and 1st generation Somali-Australian women in Australia [32, 43, 45]. Third, fasting and giving alms to the poor, as a conduit for healing—among Somali immigrants living with diabetes in Sweden [52].

In some studies, participants stressed the importance of combining religious faith and biomedical treatment. In one US-based study on mental health within Somali communities, Michlig et al (2021) [37] observed that participants believed in the value of "both prayer and medical support. . .during times of injury of disease" (p.6). This belief was expressed through the Somali proverb: "you should trust your God, and tie your camel." (p.6.)

*"about the way I care of my diabetic, number one is God. He [God] is the one that is giving me direction, helping me out, it is not my power. It is God's power. It's God's power that is helping me get through this diabetic. Thank God for that" (P6, in Kindarara, 2017 [29])*

*If I get sick my Somali friends will visit me and pray for me,. . . Somalis we strongly believe that Quran is the best treatment and if you believe it, other modern medicine will work. (Dalmar, Somali man in Kokanovic, 2008 [32])*

*It doesn't matter whether it's mental or regular hospital, we read Qur'an on the person. We have been told to do that. For example you take flowers. For us, we read Qur'an."* [22].

*3.1.3.3 Traditional medicine and care practices.* In the Netherlands, Ghanaians living with hypertension reported using herbal medications in treating hypertension [42]. Knowledge of these medicines was drawn from family and social networks, mass media (local community radio) and travels home.

*When they first told me, I was worried..., some friends told my husband about some Ghana medicine that helped them... . I bought some and my husband went to Ghana and brought some for me to drink like tea... . herbs and moringa and they said I should add ginger and garlic. . . .I took it with the medicine... it was natural, so I didn't inform him [doctor].. ." IDI-Female– 08 in Nyaaba, 2019* [42]*)*

*You know we listen to radio on the [inter]net. . .we hear about the medicines from the Ghana FMs... oh, you family member can buy and send to you if you send the money (IDI-Male-01in Nyaaba, 2019* [42]*)*

Somali refugee participants in the US reported that individuals with severe mental illness were managed with culturally-informed care practices at home–therapeutic sessions of 'talking and listening' by trusted family members [22]. These individuals received constant care at home but were not allowed to leave home due to fear of causing harm to others.

*We never let that person outside because, when he goes in public, he looks differently and other people will look at that person differently. And then person will get more crazy. We make sure always we keep him at home until he gets better. . . . We make sure that he gets everything and we keep them at home as long as we can and try our best."*[22]

*Just we talk and doctor he's helping with some medicine for sleep, always he [is] quiet. And if that family has many kids, we move the kids, maybe [to] a cousin or relative, they don't have any kids or maybe they have only one or two. . . . And we also take time to talk to him when he's happy, when he's angry, everything. We want to know about his mind. (Woman in Bette-mann, 2015* [22]*)*

Three studies conducted in Switzerland [34], France [46] and the UK [54] also showed that participants' local social networks played a major role in illness action. Eritrean Asylum-seekers in Switzerland depended on their local social networks for providing care for members with mental health problems as they faced difficulties accessing formal mental health care services [34].

*The role of other people in the mental health of someone... has a great role, basically, especially in conditions like Eritrea, we are very connected. We have a connected culture [. . .of]helping each other. It's not like leaving someone alone or isolating them. So, this kind of living jointly together and harmonizing culture, it helps you a lot, basically. (Male, age 30–35 years, Mel-amed et al., 2019* [34]*).*

*3.1.3.4 Transnational therapy networks and 'therapeutic journeys'.* Omar et al (2017) used the concept 'therapeutic journeys' to describe how some Somali and Eritrean men sought solutions for the mental health problems from their home countries [43]. Elderly men dealt with

emotional difficulties by going back home to find relief while younger men were repatriated by their parents to undergo identity and mental health strengthening "cultural rehabilitation".

> *Especially amongst young men who have drug abuse or mental health issues . . . there's a growing trend that families send their young one back home for cultural rehabilitation . . . . They will live a better standard; they will be . . . regarded as high class when they go back . . . . I think that helps them with their self-esteem . . . (Community representative in Omar et al., 2017* [43])

We identified two types of 'therapeutic journeys' across a selection of studies. Actual physical journeys home as described by Omar et al (2017) [43]; and 'courtesy journeys' whereby relatives and friends brought back medicines and other therapeutics aids from their travels home. Both types of journeys were facilitated by transnational therapy networks [10, 11] and mediated by what Gastaldo et al (2004) [59] cited by Majeed (2021, p. 2) [60] termed 'therapeutic landscapes of the mind'–the influence of place-based memories on healthcare choices for immigrant communities.

For West Africans in the UK, health problems were thought through via the lenses of home [18]; in Somali-Australian communities parents sent young children with behavioural and mental health conditions home [45].

> "I feel like a lot of the kids that are taken back home (dhaqan ceelis), there might be something wrong (mental health issues), but people (parents and older generation) do not realise." (FGD 2, Participant, Said et al, 2021,p.934 [45])

## Discussion

We discuss the major findings and insights along the three thematic areas we explored: meanings, experiences and illness action.

### Meanings of chronic conditions

In studies that explored meanings there was awareness and knowledge of major chronic conditions—hypertension, diabetes, stroke, cancers and mental illness (in particular depression). Participants across several studies also demonstrated shared understandings of the major risk factors of these chronic diseases, such as poor diets, physical inactivity, alcohol overconsumption, and psychosocial stress. Some conditions, like prostate cancer, were less well known–knowledge of these conditions coincided with a personal diagnosis, or the diagnosis of a significant other.

Chronicity has been defined as "trajectories of *self-care, caregiving and health care, as health, illness and disease co-exist and co-evolve* in the setting of primary care, local care networks (within families and communities) and at times institutions (including traditional healthcare systems)" [61]. There was a shared understanding of this concept across a number of study communities. For many, chronic conditions were life-long conditions that required complex interventions across the life course. In a few studies, some participants held beliefs about the curability of chronic conditions (and denials about the long-term impact of chronic diseases). Within this cohort of participants, these ideas were held at early stages of chronic illness experiences; and were abandoned as time and the demands of care progressed. The scope of knowledge of conditions and their risk factors was associated with educational status, lived experience of conditions or of caregiving, and access to biomedical healthcare professionals.

Culture mediated knowledge in areas where chronic conditions were attributed to supernatural causes, and also in instances where individuals compared drivers of disease 'back home' in their countries of origin, to drivers of disease in their countries of settlement. Food, dietary habits, stress, and quality of social support for example, were dominant comparative examples.

Individuals made sense of new diagnoses and evolving illness experiences by gathering information from eclectic sources: hospital settings, friends, local communities, religious institutions, media (traditional and social) and families at home and abroad. Some participants also drew knowledge from their lived experience. As has been reported in studies on diabetes and hypertension experiences across cultures [6, 15], participants 'body-listened' and made illness action choices based on their embodied knowledge. These lay rationalities, forged in multi-cultural and multi-institutional spaces, as well as embodied experiences, were dynamic and functional. They could also be contradictory. Mapping meanings and sources of knowledge highlighted the 'cognitive polyphasic' [8] character of lay understandings of chronic conditions. Conditions were associated simultaneously with medical, social, structural and supernatural factors, for example for mental illness among Somali and Ethiopian communities in Australia and the US [22, 32], and for diabetes among East African communities in the UK [25].

## Experiences

The disruptive impact of chronic conditions was the major focus of the majority of studies. Depending on the condition and length of chronic illness experience, disruption occurred across several domains of everyday life–physical, sexual, psychological, social and financial. For the more marginalised communities, such as recent refugee and asylum seeking groups, several disruptions intersected. Somali and Ethiopian women living with chronic pain in Denmark and Norway for example spoke of the intersecting problems of financial insecurity and social isolation [36, 41]. For refugee communities in Australia and the US structural barriers (such as immigration policies) were perceived as both drivers of mental health problems and risk factors for prolonged psychological distress–these barriers reinforced other disruptions such as family separation and marital tensions (e.g Asgary, 2011 [20]; Kokanovic, 2008 [32]).

The concept of chronicity allows for both positive and negative experiences–good days and bad days along the chronic illness trajectory [62]. The majority of studies focused on the more challenging and disruptive aspects of illness experiences; the positive aspects were not explicitly explored. However, patient narratives in some studies provided useful insights. For instance having a family history of a chronic condition, or living with a significant other who had a chronic condition (a spouse or relative) created conditions for healthier engagement with drug management, dietary and self-care practices. These insights emerged in studies with Ghanaian communities living with hypertension and diabetes in the Netherlands [42] and East African communities affected by a range of chronic conditions in the UK [25].

## Illness action

Illness action has been defined as strategies adopted, and resources mobilised, by individuals with chronic conditions and their significant others to manage their conditions over time [8]. Four types of illness action emerged across the studies: biomedical treatment, faith-based action, traditional treatment, and healershopping through transnational therapeutic journeys.

In the majority of studies participants drew on material resources (advice, money, social support) as well as symbolic resources (emotions, faith). Illness action strategies were often collaborative, thought through and supported through local care networks (spousal, family and social networks).

For example, a young Nigerian man with type 2 diabetes modified his traditional Nigerian diet—implicitly perceived as unhealthy—with the help of a British-Nigerian friend who liked English food–implicitly perceived as healthy [17]. Or a young Londoner with sickle cell disease negotiated in-patient care on their own terms, based on their knowledge of hospital cultures and embodied experience [33]. When Ghanaians in the Netherlands with hypertension or Somali men in Australia with mental health problems took therapeutic journeys home to Ghana or Somalia, they drew on transnational therapy networks of family, friends and socially approved health experts [11].

Gastaldo et al (2004) cited by Majeed (2021, p. 2) [60] used the term 'therapeutic landscapes of the mind' to describe the way immigrants use place-based memories—of what is healthy or generates well-being in countries of origin—to support their mental health and well-being in their country of settlement. This socio-cognitive process underpinned the two types of therapeutic journeys identified in the studies–in the way risk factors were perceived through enculturation processes, and the pull of 'home' (ie country of origin) solutions, broadcast and shared via media, social and family networks, shaped illness action intentions and practices.

Some studies highlighted ideal illness action when participants had access to affordable compassionate care in healthcare institutions and community health centres (for example for individuals living with mental health problems in the UK and Australia), and to sustained forms of social support (such as among Somali and Ethiopian communities in the US and Australia).

Multi-level barriers to illness action were also reported across several studies. At the individual level barriers included the physical impact of a condition (e.g chronic pain), psychological challenges regarding changing habits (eating different foods), managing stigma (perceived, actual, courtesy), and managing family, social and financial challenges. At the inter-personal level relational tensions, both real and perceived, undermined the ability for the chronically ill to share their problems and seek support. At the family and community level some forms of psychosocial support reinforced stigmatising attitudes and outcomes, such as keeping individuals living with mental health conditions at home with minimal access to social life. At the structural level, prejudice, racism and biases undermined healthcare service provision in hospitals. For refugee and asylum-seeking groups anti-immigration policies were reported to cause psychosocial stress and undermine health outcomes, particularly for those living with mental health problems. These structural barriers were reported among participants in the UK, Netherlands, Australia and the US.

There was nuanced awareness, dynamism and contradiction in narratives on illness action. For conditions like schizophrenia and sickle cell disease, the quality of support in the private sphere (of family and significant others) was rated higher than support in the public sphere of health and social care services. However, both spheres, as participant narratives showed, were also contested spaces: disease stigma in communities and racial discrimination in healthcare settings, lead to psychological distress and strategies of disclosure that could undermine health outcomes (e.g Michlig et al, 2022 [37]; Said et al 2021 [45]).

In research on African communities, culture is often held up as a barrier to ideal healthcare choices [8]. In a number of studies cultural attitudes and cultural practices had a health disabling impact, on defining health problems (e.g drawing on supernatural causal theories), managing chronic conditions (e.g committing to healthy diets) and negotiating social support (e.g dealing with self-stigmatising practices within communities or gender-based tensions around social roles and responsibilities). But the positive side of culture was also reported such as family support for chronically ill individuals, and significant others facilitating access to indigenous medicines or healing services. For those who could embark on therapeutic

journeys, the psychosocial impact of going 'back home' lessened the burdens of negotiating the difficulties of life in their countries of settlement.

## Limitations

Methodological and interpretive limitations emerged in some studies. Some studies replicated some of the limitations in healthcare for minoritized ethnic groups—for example conducting research in English exclusively. For individuals who were more fluent in their mother tongues, this approach erased nuanced meanings and understandings, very similar to the ways communication between patients and healthcare providers was reported to be undermined by linguistic barriers. Some studies also took participants' analysis of healthcare issues at face value–for example not questioning erroneous ideas about prevalence rates of chronic conditions in African countries, or essentialist notions of representations of chronic conditions in African communities. While qualitative methods privilege the lived experiences and perspectives of participants, critical approaches create the conditions for productive dialogue and conscientization. This allows for deeper understandings of themes under exploration between researchers and research participants. The critical approach is particularly important for conducting research with minoritized ethnic groups whose lives and experiences are often marginalised in public health research in the Global North [8].

Finally, there was very limited research on individuals of African heritage who were born in the study countries. This omission was particularly problematic in the UK and the US where African communities have had established generational roots for at least a century (e.g Adi, 1998 [63]; Falola, 2013 [64]) and where a growing body of work examines the lived experiences of individuals with 'hyphenated identities' [65]. There are qualitative differences between British or US born individuals of African heritage (who may identify as British-Ghanaian or Nigerian-American) and recent African immigrants to the UK or US–not only in terms of identities, but also in terms of psychosocial experiences, and health and treatment seeking behaviours within formal healthcare institutions. Said et al (2021) [45] for example highlighted inter-generational differences in strategies of disclosure; this is important because fear of disclosure is associated with stigma consciousness and this phenomenon is shared across national and cultural groups and informs decision making on illness action.

## Conclusions

Cumulatively, the studies explored perspectives of 748 participants, from 27 African countries (including a minority born in the study countries), living with or affected by 8 conditions. The numbers were not sufficient to make generalizable conclusions about the broader communities of which they are members. A focus on nationality (e.g Ghanaian, Nigerian) solely, commits a 'fallacy of homogeneity' [66] by obscuring ethnic and linguistic heterogeneity within these national groups–for example in Ghana, 40 languages are spoken across 8 major ethnic groups [67]. Secondly, studies focused on a limited set of eight chronic conditions–type 2 diabetes, hypertension, prostate cancer, sickle cell disease, chronic hepatitis, chronic pain, musculoskeletal orders and mental health conditions. Our search did not yield any studies on experiences of neurodegenerative conditions, for example, although prevalence rates of these conditions are rising in ageing groups in African communities on the continent and the Global North [68]. Within the reviewed studies themselves participants listed more prevalent conditions in their communities, beyond the eight in focus.

However, the majority of studies outlined implications for research, practice and policy that were broadly shared. Recommendations included improving public health education (increasing awareness of a broader range of chronic conditions; addressing disease stigma, especially

mental illness); investing in accessible, ethical and socially just healthcare (diagnosis, treatment, language barriers, cultural competence); and strengthening community-based care addressing social factors such as language barriers. Some studies recommended more critical conceptual approaches to researching and addressing the relationship between structural inequalities and poor health in African diaspora communities [37].

These findings present an important start to examining a much-ignored area that intersects healthcare for African communities in the Global North, and pluralistic healthcare on the African continent and raises important conceptual, methodological and policy challenges for national health programmes on healthcare disparities as well as for global health.

## Supporting information

**S1 Table. PRISMA checklist.**
(DOCX)

**S2 Table. Search strategy.**
(DOCX)

**S3 Table. COREQ-32 checklist.**
(DOCX)

## Author Contributions

**Conceptualization:** Ama de-Graft Aikins.

**Data curation:** Ama de-Graft Aikins, Olutobi Sanuade, Leonard Baatiema, Kafui Adjaye-Gbewonyo.

**Formal analysis:** Ama de-Graft Aikins, Olutobi Sanuade, Leonard Baatiema, Kafui Adjaye-Gbewonyo, Juliet Addo, Charles Agyemang.

**Writing – original draft:** Ama de-Graft Aikins, Olutobi Sanuade, Leonard Baatiema.

**Writing – review & editing:** Ama de-Graft Aikins, Olutobi Sanuade, Leonard Baatiema, Kafui Adjaye-Gbewonyo, Juliet Addo, Charles Agyemang.

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
