## [Decision Letter · Decision Letter 0]

2 Oct 2022

PONE-D-22-22810How chronic conditions are understood, experienced and managed within African communities in Europe, North America and Australia: a synthesis of qualitative studiesPLOS ONE

Dear Dr. Aikins, 

Thank you for submitting your manuscript to PLOS ONE. After careful consideration, we feel that it has merit but does not fully meet PLOS ONE’s publication criteria as it currently stands. Therefore, we invite you to submit a revised version of the manuscript that addresses the points raised during the review process.

We look forward to receiving your revised manuscript.

Kind regards,

Donovan Anthony McGrowder, PhD., MA., MSc

Academic Editor

PLOS ONE

Journal Requirements:

2.We note that this manuscript is a systematic review or meta-analysis; our author guidelines therefore require that you use PRISMA guidance to help improve reporting quality of this type of study. Please upload copies of the completed PRISMA checklist as Supporting Information with a file name “PRISMA checklist”.

Additional Editor Comments:

Dear Dr. Aikins,   

Your manuscript “How chronic conditions are understood, experienced and managed within African communities in Europe, North America and Australia: a synthesis of qualitative studies” has been assessed by our reviewers. They have raised a number of points which we believe would improve the manuscript and may allow a revised version to be published in PLOS ONE. Their reports, together with any other comments, are below.

If you are able to fully address these points, we would encourage you to submit a revised manuscript to PLOS ONE.

Regards,

Dr. Donovan McGrowder

Associate Editor

Reviewers' comments:

Reviewer's Responses to Questions

**Comments to the Author**

1. Is the manuscript technically sound, and do the data support the conclusions?

Reviewer #1: Yes

Reviewer #2: Yes

2. Has the statistical analysis been performed appropriately and rigorously? 

Reviewer #1: I Don't Know

Reviewer #2: Yes

3. Have the authors made all data underlying the findings in their manuscript fully available?

Reviewer #1: Yes

Reviewer #2: Yes

4. Is the manuscript presented in an intelligible fashion and written in standard English?

Reviewer #1: Yes

Reviewer #2: Yes

5. Review Comments to the Author

Reviewer #1: This review focuses on the lived experiences of chronic conditions among African

communities in the Global North, focusing on established immigrant communities as well as

recent immigrant, refugee, and asylum-seeking communities. We conducted a systematic and

narrative synthesis of qualitative studies published from inception to 2022, following a search

from nine databases - MEDLINE, EMBASE, PsycINFO, Web of Science, Social Science

Citation Index, Academic Search Complete, CINAHL, SCOPUS and AMED. 39 articles

reporting 32 qualitative studies were included in the synthesis. The studies were conducted in

10 countries (Australia, Canada, Denmark, France, Netherlands, Norway, Sweden,

Switzerland, United Kingdom, and the United States) and focused on 748 participants from

27 African countries. The majority of participants believed chronic conditions to be lifelong,

requiring complex interventions. Chronic illness impacted several domains of everyday life -

physical, sexual, psycho-emotional, social, and economic. Participants managed their illness

using biomedical management, traditional medical treatment and faith-based coping, in

isolation or combination. In a number of studies, participants took ‘therapeutic journeys’ –

which involved navigating illness action at home and abroad, with the support of

transnational therapy networks. Multi-level barriers to healthcare were reported across the

majority of studies.

Overall, an interesting contribution. MINOR COMMENTS:

1. Please shorten the Introduction.

2. Please provide a formal concluding paragraph.

Reviewer #2: The manuscript is well-written and scientifically-sound. The authors conducted a review to examine the lived experiences of chronic conditions among African communities in Europe and North America, and other countries in the Global North, focusing on established immigrant communities as well as recent immigrant, refugee and

asylum seeking communities. We were interested in meanings, experiences and illness action

from the perspective of individuals living with chronic conditions. The study is novel and will add to literature.

---

## [Author Response · Author response to Decision Letter 0]

18 Oct 2022

RESPONSES TO REVIEWERS

Reviewer #1: This review focuses on the lived experiences of chronic conditions among African communities in the Global North, focusing on established immigrant communities as well as recent immigrant, refugee, and asylum-seeking communities. We conducted a systematic and narrative synthesis of qualitative studies published from inception to 2022, following a search from nine databases - MEDLINE, EMBASE, PsycINFO, Web of Science, Social Science Citation Index, Academic Search Complete, CINAHL, SCOPUS and AMED. 39 articles reporting 32 qualitative studies were included in the synthesis. The studies were conducted in 10 countries (Australia, Canada, Denmark, France, Netherlands, Norway, Sweden, Switzerland, United Kingdom, and the United States) and focused on 748 participants from 27 African countries. The majority of participants believed chronic conditions to be lifelong, requiring complex interventions. Chronic illness impacted several domains of everyday life - physical, sexual, psycho-emotional, social, and economic. Participants managed their illness using biomedical management, traditional medical treatment and faith-based coping, in isolation or combination. In a number of studies, participants took ‘therapeutic journeys’ – which involved navigating illness action at home and abroad, with the support of transnational therapy networks. Multi-level barriers to healthcare were reported across the majority of studies.

Overall, an interesting contribution. 

Response to general comment:

We thank the reviewer for describing our review as an “interesting contribution” to the literature.

MINOR COMMENTS:

1. Please shorten the Introduction.

Response to Comment 1: 

We have moved section on cognitive polyphasia and enculturation in the introduction (paragraph 3, page 1 in the original paper) to the Aims section on page 2, just before we outline our research questions. This provides a clearer conceptual framework for the review questions. 

2. Please provide a formal concluding paragraph.

Response to Comment 2: 

We have restructured the final paragraph (in the original paper) to incorporate a formal concluding paragraph (page 38).

Reviewer #2: The manuscript is well-written and scientifically-sound. The authors conducted a review to examine the lived experiences of chronic conditions among African communities in Europe and North America, and other countries in the Global North, focusing on established immigrant communities as well as recent immigrant, refugee and

asylum seeking communities. We were interested in meanings, experiences and illness action

from the perspective of individuals living with chronic conditions. The study is novel and will add to literature.

Response to general comment: We thank Reviewer 2 for describing our review as a ‘novel’ addition to the literature.

---

## [Editor Report · Decision Letter 1]

25 Oct 2022

How chronic conditions are understood, experienced and managed within African communities in Europe, North America and Australia: a synthesis of qualitative studies

PONE-D-22-22810R1

Dear Dr. Aikins,

We’re pleased to inform you that your manuscript has been judged scientifically suitable for publication and will be formally accepted for publication once it meets all outstanding technical requirements.

Kind regards,

Donovan Anthony McGrowder, PhD., MA., MSc

Academic Editor

PLOS ONE

Additional Editor:

Dear Dr. Aikins,

The manuscript entitled “How chronic conditions are understood, experienced and managed within African communities in Europe, North America and Australia: a synthesis of qualitative studies” was revised in accordance with the reviewers’ comments and is provisionally accepted pending final checks for formatting and technical requirements.

Regards,

Dr. Donovan McGrowder (Academic Editor)

---

## [Editor Report · Acceptance letter]

2 Nov 2022

PONE-D-22-22810R1 

How chronic conditions are understood, experienced and managed within African communities in Europe, North America and Australia: a synthesis of qualitative studies 

Dear Dr. de-Graft Aikins:

I'm pleased to inform you that your manuscript has been deemed suitable for publication in PLOS ONE. Congratulations! Your manuscript is now with our production department. 

Kind regards, 

on behalf of

Dr. Donovan Anthony McGrowder 

Academic Editor

PLOS ONE